# Electropolymerized Aniline-Based Stainless Steel Fiber Coatings Modified by Multi-Walled Carbon Nanotubes for Electroanalysis of 4-Chlorophenol

**DOI:** 10.3390/ma15103436

**Published:** 2022-05-10

**Authors:** Krzysztof Kuśmierek, Katarzyna Skrzypczyńska, Andrzej Świątkowski, Ewa Wierzbicka, Izabella Legocka

**Affiliations:** 1Institute of Chemistry, Military University of Technology, 00-908 Warsaw, Poland; krzysztof.kusmierek@wat.edu.pl (K.K.); a.swiatkowski@wp.pl (A.Ś.); 2Department of Polymer Technology and Processing, Łukasiewicz-Industrial Chemistry Institute, 01-793 Warsaw, Poland; katarzyna.skrzypczynska@ichp.lukasiewicz.gov.pl (K.S.); izabella.legocka@ichp.lukasiewicz.gov.pl (I.L.)

**Keywords:** electropolymerization, polyaniline, multi-walled carbon nanotubes, differential pulse voltammetry, 4-chlorophenol

## Abstract

In this paper, a stainless steel fiber coated electropolymerized aniline, without and with carbon nanotubes (SS/PANI and SS/PANI/CNT), along with CNTs modified carbon paste electrodes (CPEs), were prepared. The electrodes were characterized by differential pulse voltammetry (DPV) and applied for the detection of 4-chlorophenol (4-CP). For all the electrodes, the oxidative peak current showed a linear dependence on the 4-CP concentration in the range of 0.05–0.5 mmol/L with *R*^2^ ≥ 0.991. SS/PANI/CNT electrodes showed greater sensitivity for the detection of the 4-CP than the SS/PANI and CPEs. For all of the aniline-based stainless steel electrodes, both the LOD and LOQ decreased with the increase in the number of electropolymerization cycles. The lowest LOD (0.38 µmol/L) and LOQ (1.26 µmol/L) were observed for the SS/PANI/CNT electrode modified in aniline solution during 30 cycles. The methods were successfully applied to the analysis of 4-CP in real samples (tap water and river water). The results demonstrated the good agreement of the added and found concentrations of the 4-CP. The recovery and precision were from 95.12% to 102.24% and from 1.53% to 6.79%, respectively. The proposed electrodes exhibited acceptable reproducibility, admirable stability, and adequate repeatability and showed potential for the analysis of 4-CP in water.

## 1. Introduction

Polar compounds are some of the most problematic pollutants found in the aquatic environment. Among the polar compounds that have a significant impact on the environment, phenolic compounds are an important group. These compounds enter the environment either directly, as industrial effluents, or indirectly, as transformation products of natural and synthetic chemicals, including pesticides. The U.S. Environmental Protection Agency has compiled a list of 11 phenolic compounds considered to be major pollutants, among which chlorophenols are the most toxic and carcinogenic. Among the many removal methods, electrochemical hydrodehalogenation (ECH) was regarded as a promising approach, owing to its easy operations, wild reaction conditions, and high efficiency.

Polyaniline (PANI) is one of the conducting polymers; it attracts great attention due to its low cost, ease of synthesis, good environmental stability, reversible redox activity, and potential applications in sensing, energy conversion, and storage [1,2,3,4,5]. Due to its high conductivity and good environmental stability [6], as well as its ease of preparation, polyaniline is a very popular coating material for various applications [6,7,8,9,10,11]. The first paper on this subject was published in 1962 by Mohilner et al. [12], who studied the kinetics of the electrochemical oxidation of aniline in sulfuric acid. They found that initial oxidation requires the formation of an aniline radical. Since then, the nature of intermediates in PANI synthesis have been proposed based on cyclic voltammetry. The first stage of PANI synthesis was found to be comprised by the formation of the C_6_H_5_-NH_2_^+^ cation radical through the transfer of an electron from an sp^3^ orbital of a nitrogen atom to the electrode during electrochemical oxidation. Electropolymerization was carried out by voltammetry methods in a three-electrode system using a potentiostat/galvanostat. Sorptive coatings were prepared from aqueous solutions of aniline, containing added electrolyte, with H_2_SO_4_ used as the acid. Stainless steel fibers applied as stationary phases served as working electrodes. The saturated calomel electrode was used as a reference electrode, and the platinum electrode served as an auxiliary electrode. Constant potential, constant current, and potential sweep methods were usually used for the electropolymerization of PANI and its derivatives [12]. Since polymerization can only proceed when the monomer is oxidized on the electrode, these procedures are advantageous from the standpoint of controlling the amount of product. Standard electrochemical techniques using a split cell containing a working electrode, a counter electrode, and a reference electrode usually produce the required films. The most commonly used working electrodes are conductive materials such as gold-, carbon-, platinum-, and indium tin oxide-coated glass plates. Intrinsically conducting organic polymers, such as polyaniline, polythiophene, or polypyrrole, have been studied intensively during the last two decades [12].

Multi-walled carbon nanotubes (CNTs) have unique physicochemical properties, such as high surface area, ordered structure with high aspect ratio, high mechanical strength, and ultralightweight, as well as high electrical and thermal conductivity [13,14]. The combination of all these features makes CNTs very interesting materials with potential for a variety of applications. Due to their low resistivity, good electronic properties, high stability, and large available surface area, CNTs seem to be excellent fillers for polymers in regard to improving their electric conductivity, as well as their mechanical properties [15,16,17].

In this paper, the application of multi-walled carbon nanotubes in the preparation of stainless steel fiber coated with PANI subsequently used as electrodes or carbon paste electrodes was investigated. The prepared SS/PANI and SS/PANI/CNT electrodes were applied for the detection of 4-CP by differential pulse voltammetry (DPV). The capability of these newly modified stainless steel electrodes to detect chlorophenol was compared to the carbon paste electrodes modified with CNTs. To the best of our knowledge, the electrochemical determination of 4-CP at the stainless steel electrode site has not been reported so far. The electrodeposited films of polyaniline on stainless steel wires as the working electrode were investigated and characterized using a simple cyclic voltammetry method to obtain new information about the electrochemical properties of the deposited films. For potential new applications in electroanalysis, polyaniline, as well as composites of carbon nanotubes–polyaniline, have been studied. It can be expected that such electrodes can find many applications in electroanalysis for the determination of many other analytes.

## 2. Materials and Methods

### 2.1. Reagents and Materials

The 4-chlorophenol (4-CP) was obtained from Sigma-Aldrich (St. Louis, MO, USA). The hydrochloric acid, sulfuric acid, sodium sulfate, and aniline were obtained from Chempur (Piekary Śląskie, Poland). Stock standard solution of 4-CP (0.5 mmol/L) was prepared in 0.1 mol/L Na_2_SO_4_. Working solutions were prepared as required by dilution with sodium sulfate of the same concentration. The multi-walled carbon nanotubes CNTs (CNT Co., Ltd., Incheon, Korea) and graphite (<45 μm) from Sigma-Aldrich (St. Louis, MO, USA) were used. Paraffin oil was received from Fluka (Buchs, Switzerland).

### 2.2. Characterization of the CNTs

As reported elsewhere [18,19,20,21,22], the electrochemical oxidation of 4-chlorophenol at the CPEs was controlled by adsorption, and the electrode peak currents were positively correlated with the physicochemical properties of the modifier used—peak currents were increased with the specific surface area and the adsorption capacity of the modifier. Thus, the textural, as well as the adsorptive properties of the CNTs toward 4-CP, were determined.

The pore structure of the CNTs was characterized by N_2_ adsorption-desorption at 77 K using a ASAP 2020 (Micromeritics, Norcross, GA, USA) surface analyzer, while the adsorptive properties of the CNT were studied as a function of time in batch adsorption experiments.

The adsorption experiments were carried out at 25 °C in Erlenmeyer flasks containing 5 mL of the 4-CP aqueous solution (0.5 mmol/L) and 5 mg of CNTs. The samples were shaken, withdrawn at appropriate time intervals, filtered, and analyzed by UV-spectrophotometry. The amount of adsorption at the time t, *q_t_* (mmol/g), was calculated by the Equation (1):(1)$$q_{t}=\frac{\left(C_{0}-C_{t}\right)V}{m}$$
where *C*_0_ and *C_t_* are the initial and the concentration of 4-CP at the time t (mmol/L), *V* is the volume of the solution (L), and *m* is the mass of the CNTs (g).

All the experiments were carried out in duplicate and the averaged values were used for further calculations.

### 2.3. Preparation of the PANI Coated Stainless Steel Fibers

In a flask of 250 mL, a 0.2 mol/L aniline solution in 0.5 mol/L sulfuric acid was prepared. A total of 20 mL of this solution was placed in a measuring cell and then formed into a conventional three-electrode system. That system was used for all electrochemical experiments, with a stainless steel fiber (SS/PANI or SS/PANI/CNT) as a working electrode, a platinum (Pt) wire as an auxiliary electrode, and a saturated calomel electrode as a reference electrode. Stainless steel fibers of 6 cm in length were thoroughly washed following the method suggested by Masoumi et al. [23]. According to this method, the fibers were purified with acetone by placing them in an ultrasound bath for 30 min. Then they were subsequently washed twice with distilled water. Prepared fibers were stored in a desiccator at room temperature.

The fiber coatings containing polyaniline (PANI) and polyaniline with carbon nanotubes (PANI/CNT) were electrodeposited on the prepared fiber’s surface using the cyclic voltammetry technique. Before electrochemical polymerization, the 1.5 cm piece of stainless steel wire was immersed in an aqueous solution containing 0.2 mol/L of aniline and 0.5 mol/L of sulfuric acid, then deoxidized by passing through nitrogen gas for 10 min. The potential scan was cycled 10, 20, or 30 times between 0.1 V and 1.0 V vs. SCE at a scan rate of 25 mV/s. For the preparation of the PANI/CNT fiber, the same method was used; the only difference was the addition of 0.02% of multi-walled carbon nanotubes to the aqueous solution of aniline. This optimum amount of carbon nanotubes was chosen based on the literature data [13]. Then, 6 mg CNT was added, and the obtained suspension was mixed using ultrasound. The obtained fiber coatings were washed with deionized water and kept in a desiccator for 24 h at room temperature.

### 2.4. Carbon Paste Electrodes Preparation

The carbon paste electrode was prepared by thoroughly mixing graphite powder and paraffin oil (9:1 *m*/*m*) in a mortar with a pestle. The carbon paste was modified by adding 2.5, 5, or 10% (*m*/*m*) of CNTs. The modified CPEs were prepared using the same methods by precisely mixing weighed amounts of multi-walled carbon tubes with graphite powder and paraffin oil. The prepared mixtures were stored for three days in a desiccator at 25 °C. The pastes were then packed into a 3 mm diameter electrode cavity (2 mm). Before each use, the electrode’s surface was rubbed with a piece of paper until a smooth surface was obtained.

### 2.5. Voltammetry

The cyclic voltammetry (CV) and differential pulse voltammetry (DPV) measurements were performed with a three-electrode system, including the SS/PANI or SS/PANI/CNT fiber as the working electrode, or—for the sake of comparison—the carbon paste electrodes modified with CNT (CPE/CNT), a platinum wire as a counter electrode, and saturated calomel electrode as a reference electrode. Electrochemical experiments were carried out with an Autolab PGSTAT 20 potentiostat (Eco Chemie) controlled by GPES 4.9 software.

DPV tests were performed for various concentrations of 4-CP solution (in the range of 0.05 to 0.5 mmol/L) in 0.1 mol/L sodium sulfate, and the voltammograms were registered from 0 to +1.0 V at a scan rate of 50 mV/s. The pulse height and width were set as 50 mV and 50 ms, respectively, and the sampling time was 50 ms. These electrochemical conditions were applied based on the previously described results [18,19,20,21,22].

## 3. Results

### 3.1. Textural and Adsorptive Properties of the CNTs

The nitrogen adsorption/desorption isotherm for the CNTs was determined at 77 K, and the results are presented in Figure 1. The specific surface area of the CNTs was calculated using the Brunauer–Emmett–Teller equation and was found to be 195 m^2^/g. The total pore volume (V_t_), the micropore volume (V_mi_), and the mesopore volume (V_me_) were 0.75, 0.09, and 0.74 cm^3^/g, respectively.

The adsorption kinetic curve of the 4-chlorophenol (the plot of *q_t_* = *f*(*t*)) is shown in Figure 2a. The adsorption equilibrium was achieved after 60–90 min. After this period, approximately 60% of the 4-CP was adsorbed on the CNTs (*q_e_* = 0.235 mmol/g).

For the description of the experimental data, the pseudo first-order (PFO) (Equation (2)), pseudo second-order (PSO) (Equation (3)), as well as the Weber–Morris intraparticle diffusion models (Equation (4)) [23] were considered:(2)$$log\left(q_{e}-q_{t}\right)=log\;q_{e}-\frac{k_{1}}{2.303}t$$
(3)$$\frac{t}{q_{t}}=\frac{1}{k_{2}q_{e}^{2}}+\frac{1}{q_{e}}t$$
(4)$$q_{t}=k_{i}t^{1/2}+C_{i}$$
where: *q_e_* is the equilibrium amount of 4-CP adsorbed per unit mass of the CNTs (mmol/g); *k*_1_ is the rate constant of the PFO adsorption model (1/min); *k*_2_ is the rate constant of the PSO adsorption model (g/mmol·min); *k_i_* is the intraparticle diffusion rate constant (mmol/g·min^1/2^); and *C_i_* is the thickness of the boundary layer.

The PFO and PSO rate constants were 0.006 1/min and 0.946 g/mmol·min, respectively. The correlation coefficient for the PFO kinetic model was relatively low, whereas the PSO model gives a better fitting with the high *R*^2^ value (0.892 vs. 0.999). This indicates that the adsorption system belongs to the second-order kinetic model. The plot of *q_t_* vs. t^1/2^ for the intraparticle diffusion is shown in Figure 2b. As can be seen, the plot was not linear over the whole time range, and the curves do not pass through the origin. This suggests that the adsorption mechanism of the 4-CP on CNTs is complex, and both the surface adsorption, as well as the intraparticle diffusion, contribute to the adsorption process.

### 3.2. Electropolymerization of PANI Fiber Coating

The voltammetric curves obtained by electropolymerization of aniline with a wire made of stainless steel in an aqueous solution of aniline and sulfuric acid carried out at 30 cycles are shown in Figure 3.

While analyzing the voltammograms and peak currents, it can be concluded that the polymer layer formed during the first cycle affects the autocatalytic process of electropolymerization, which means that the further development of the polymer is faster than on a clean electrode. Further cyclization leads to an increase in the polymerization process. SEM images of the stainless steel fiber before and after electropolymerization were recorded using a scanning electron microscope QUANTA 3D FEG at a magnification of 100× to estimate the thickness of the layer. The resulting images are shown in Figure 4.

The obtained cover border wire polymers have an irregular structure and uneven coverage. A further coverage area is uniformly covered with a layer of polyaniline. Based on these images, the coating thickness of the produced polymer was estimated. The thickness of the clean fiber (uncoated with the polymer) was measured, and the thickness of the fiber was measured where the layer thickness appeared to be constant. The fiber diameters were then estimated using a scale, and the film thickness was calculated. The results presented in Table 1 show that as the subsequent cycles are carried out, the thickness and volume of the resulting polymer increase. In Figure 5, the SEM images of the surface of the stainless steel fiber covered with PANI/CNT layers at 10,000-fold magnification are presented.

In addition to the structure repeated in all photos, the presence of carbon nanotubes can be seen (Figure 5a). In the case of more cycles (Figure 5b,c), the carbon nanotubes were probably covered by successive layers of polyaniline. All PANI/CNT layers show a similar surface morphology. Some repeating dendrite-shaped structures are noticeable.

More detailed studies on the physicochemical, as well as the mechanistic, properties of aniline-based fiber coatings have been conducted by other authors and are reported elsewhere [8,24,25].

### 3.3. Electrochemical Studies

In the first step, the effect of accumulation time on the peak current value was determined. The accumulation time varied from 1 min to 15 min, and the corresponding current value was measured using a 0.5 mmol/L 4-CP solution. The effect of accumulation time on the 4-CP peak current is presented in Figure 6. As can be seen, the peak current increased with increasing accumulation time until about 6 min, and then stabilized. Further increasing the accumulation time did not improve the amount of 4-CP on the electrode surface, due to surface saturation. The results show that the accumulation time is not correlated with the adsorption kinetics of 4-CP on the materials used as CPE modifiers. For further experiments, an accumulation time of 7 min was chosen as the optimum for all electrodes.

Figure 7 shows the cyclic voltammograms (CV) and Figure 8 shows the differential pulse voltammograms (DPV) of 0.5 mmol/L 4-CP in 0.1 mol/L solutions of Na_2_SO_4_ obtained for SS/PANI and SS/PANI/CNT after 10, 20, and 30 electropolymerization cycles, as well as for CPEs modified by the addition of 2.5, 5, and 10% (*m*/*m*) of CNTs. For all the DPV curves, the peak currents and the peak potentials were determined. The peak potentials reveal similar values of 0.76 ± 0.01 V for all the electrodes. The peak currents in the case of each scan are about 50% higher when the number of scans increases. The increase in the modifier content yields an increase in the peak current. The results obtained indicate that the modification of the electrodes improves the performance of chlorophenol electroanalysis. It was also found that the addition of carbon nanotubes to PANI in layers applied by electropolymerization, e.g., on a wire made of stainless steel, significantly increased the possibility of using electrodes coated with the polymer in the analysis of 4-CP. The presence of the nanotubes, which increases the value of the current peaks in the curves, results in large differences in the polymer layer thicknesses. This produces a composite PANI/CNT obtained by the synergistic effect of both components: a conductive polymer and a nano-carbon.

### 3.4. Calibration and Validation

The main aim of this study was to develop sensing electrodes using aniline-based stainless steel fiber coatings, with and without multi-walled carbon nanotubes, for the electroanalysis of 4-chlorophenol. The capability of these stainless steel electrodes to detect 4-CP was compared with the CPEs modified by adding different amounts of CNTs (2.5–10%, *m*/*m*). The calibration curves of 4-CP (0.05–0.5 mmol/L) for all of the electrodes were constructed by plotting integrated peak area vs. concentration. All the measurements were carried out in triplicate and outliers were not excluded. The calibration data are summarized in Table 2.

The limits of detection (LOD) and quantification (LOQ) were calculated using Equations (5) and (6), respectively, in which SD is the standard deviation of the blank signal and *a* is the analytical curve slope [26].
(5)$$\mathrm{LOD}=\mathrm{SD}\frac{3.3}{a}$$
(6)$$\mathrm{LOQ}=\mathrm{SD}\frac{10}{a}\;$$

As can be seen, all the plots were linear in the studied range, with *R*^2^ values ≥ 0.991. In general, SS/PANI electrodes showed lower sensitivity than the CPEs and SS/PANI/CNT electrodes. For all of the aniline-based stainless steel electrodes, both the LOD and LOQ decreased with the increase in the number of electropolymerization cycles. Moreover, the SS/PANI fiber coatings modified by multi-walled carbon nanotubes showed significantly greater sensitivity for the detection of 4-CP than the CPEs. In the case of CPEs, the signal response increased with an increase in the amount of the modifier (CNTs) used. The lowest LOD (0.38 µmol/L) and LOQ (1.26 µmol/L) were observed for the SS/PANI/CNT electrode modified in aniline solution during 30 cycles. In comparison, the LOD and LOQ values obtained for the electrode prepared under the same conditions (30 cycles), but without the addition of carbon nanotubes (SS/PANI), were approximately 4.5 times higher, and were 1.72 and 5.73 µmol/L, respectively. The LOD of 4-CP obtained by SS/PANI/CNT/30 appears to be comparable, and in some cases, superior to that obtained by the other reported electrodes, as shown in Table 3.

To assess the quality of the methods (electrodes), accuracy and precision were determined. These validation tests were performed for the best three electrodes from each type—SS/PANI/30, SS/PANI/CNT/30, and CPE/CNT/10. Accuracy, intraday precision, and interday precision for 4-CP were determined by adding the standard to 0.1 mol/L sodium sulfate samples at low (0.1 mmol/L) and high concentration (0.5 mmol/L) levels. To evaluate intra-assay and inter-assay precision, five replicates for each concentration of the analyte were measured in a single run and on five consecutive days in a week, respectively. Intra-assay and inter-assay imprecision were expressed as the coefficient of variation (CV). Mean accuracies and coefficients of variation of 4-CP were presented in Table 4. The CVs of the intraday precision were in the range of 1.67–6.25% and 0.16–1.47%, while the CVs of interday precision were in the range of 5.80–9.90% and 0.82–8.26% for the initial analyte concentration of 0.1 and 0.5 mmol/L, respectively. Recoveries were within the range of 97.96% to 101.91% for intraday assay, while in the interday assay, the range of accuracy was within 97.22% to 104.50% for the initial analyte concentration of 0.1 mmol/L.

### 3.5. Analytical Application of the Electrodes

The described methods using SS/PANI/30, SS/PANI/CNT/30, and CPE/CNT/10 electrodes have been applied to determine 4-CP in tap water and Vistula River water after the addition of 0.1 mol/L sodium sulfate. Since no 4-CP was detected in the samples, the water samples were spiked with 0.2 mmol/L of the analyte. The obtained results are listed in Table 5. As can be seen, they demonstrate the good agreement of the added and found concentrations of the 4-CP. The recoveries were from 95.12% to 102.24%, and the CV was equal to or below 6.79%. This indicates that the electrodes prepared in this study should be promising sensors for the efficient determination of 4-CP.

## 4. Conclusions

In this paper, a stainless steel fiber-coated electropolymerized aniline, with and without carbon nanotubes (SS/PANI and SS/PANI/CNT), was prepared and used for the detection of 4-chlorophenol based on differential pulse voltammetry. Carbon paste electrodes modified with the same CNTs were also prepared and used. The sensitivity of the aniline-based stainless steel electrodes increased with the increase in the number of electropolymerization cycles. Even a small addition of a modifier (CNTs) to a SS/PANI electrode increases peak currents. The lowest LOD (0.38 µmol/L) and LOQ (1.26 µmol/L) were observed for the SS/PANI/CNT electrode modified in aniline solution during 30 cycles. The sensitivity of these methods is comparable to, or slightly worse, than those of electrochemical methods for 4-CP detection described by other authors. Finally, the methods were applied to the analysis of 4-CP in tap water and river water samples. All the electrodes demonstrated satisfactory results in real samples, showing a potential for the analysis of 4-CP in water.

## Figures and Tables

**Figure 1 materials-15-03436-f001:**
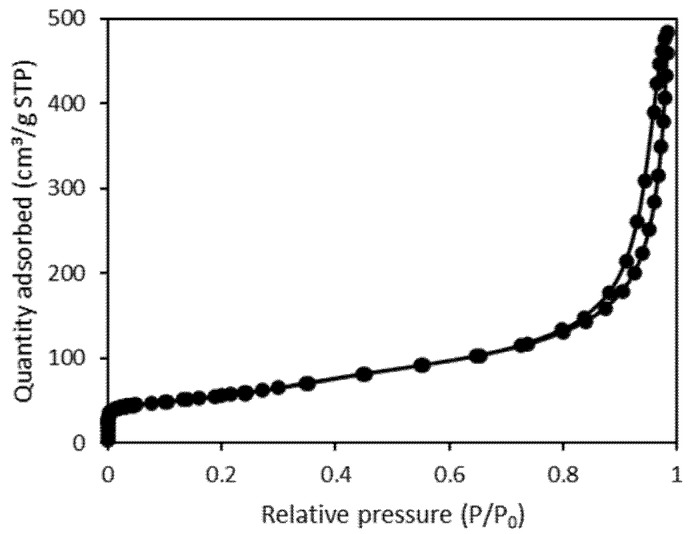
Nitrogen adsorption-desorption isotherm of CNTs at 77.4 K.

**Figure 2 materials-15-03436-f002:**
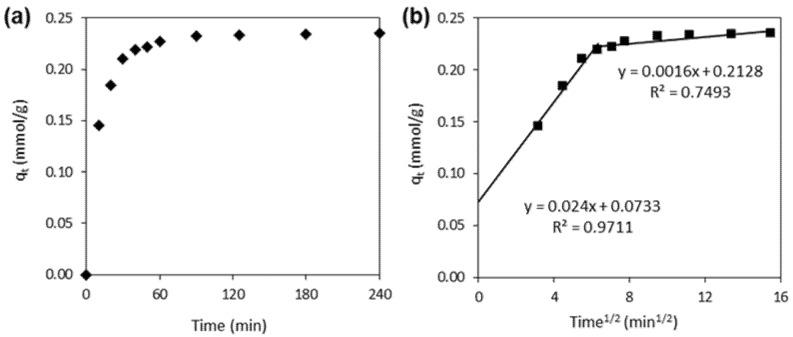
(**a**) The adsorption kinetics of 4-CP onto CNTs, and (**b**) the intraparticle diffusion kinetic model.

**Figure 3 materials-15-03436-f003:**
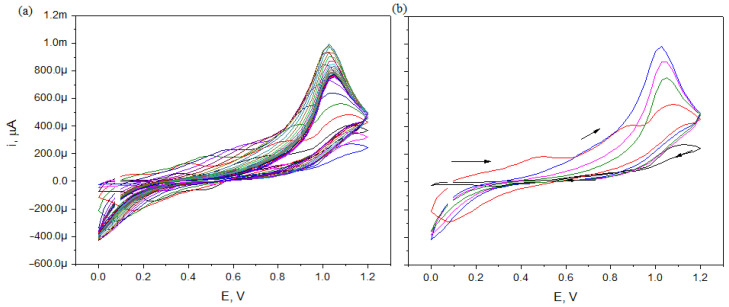
The CV curves registered for 0.2 mol/L aniline in 0.5 mol/L H_2_SO_4_ solutions using stainless steel wire electrodes after 1–30 cycles (**a**) and chosen for **----** 1, **----** 5, **----** 10, **----** 20, **----** 30 cycles (**b**).

**Figure 4 materials-15-03436-f004:**
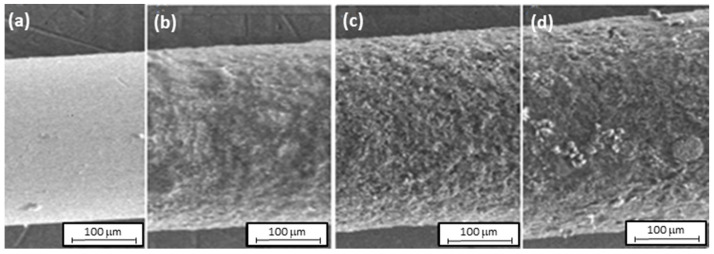
The SEM image of the stainless steel fiber before the electropolymerization (**a**) and after 10 (**b**), 20 (**c**), and 30 (**d**) cycles.

**Figure 5 materials-15-03436-f005:**
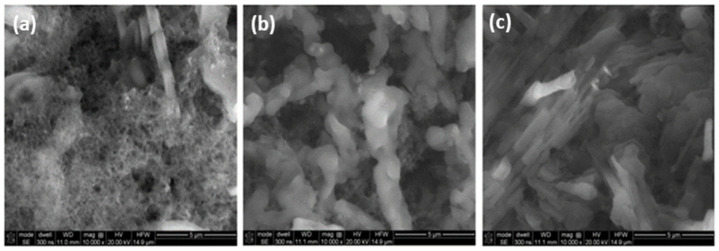
The SEM images of the stainless steel fiber coated with PANI/CNT layers after 10 (**a**), 20 (**b**), and 30 (**c**) CV cycles.

**Figure 6 materials-15-03436-f006:**
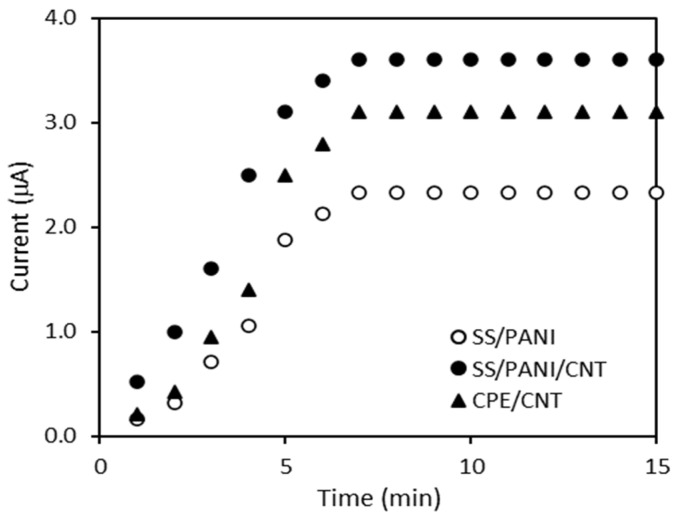
The effect of accumulation time on the peak current of the 0.5 mmol/L 4-CP in 0.1 mol/L Na_2_SO_4_ solutions.

**Figure 7 materials-15-03436-f007:**
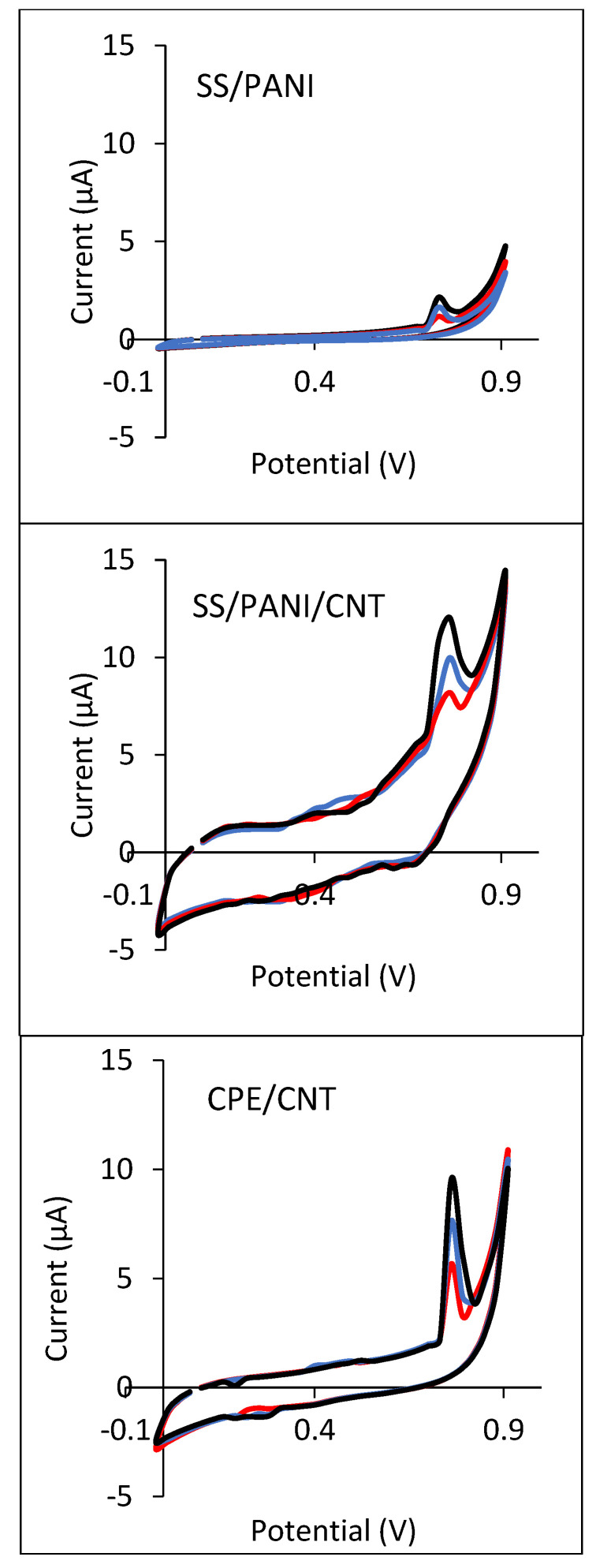
Cyclic voltammograms registered for 0.5 mmol/L 4-CP in 0.1 mol/L solutions of Na_2_SO_4_ on SS/PANI and SS/PANI/CNT electrodes obtained after 10 cycles **----**, 20 cycles **----**, 30 cycles **---**, as well as on CPEs modified with CNTs content of 2.5% **----**, 5% **----**, and 10% **---**.

**Figure 8 materials-15-03436-f008:**
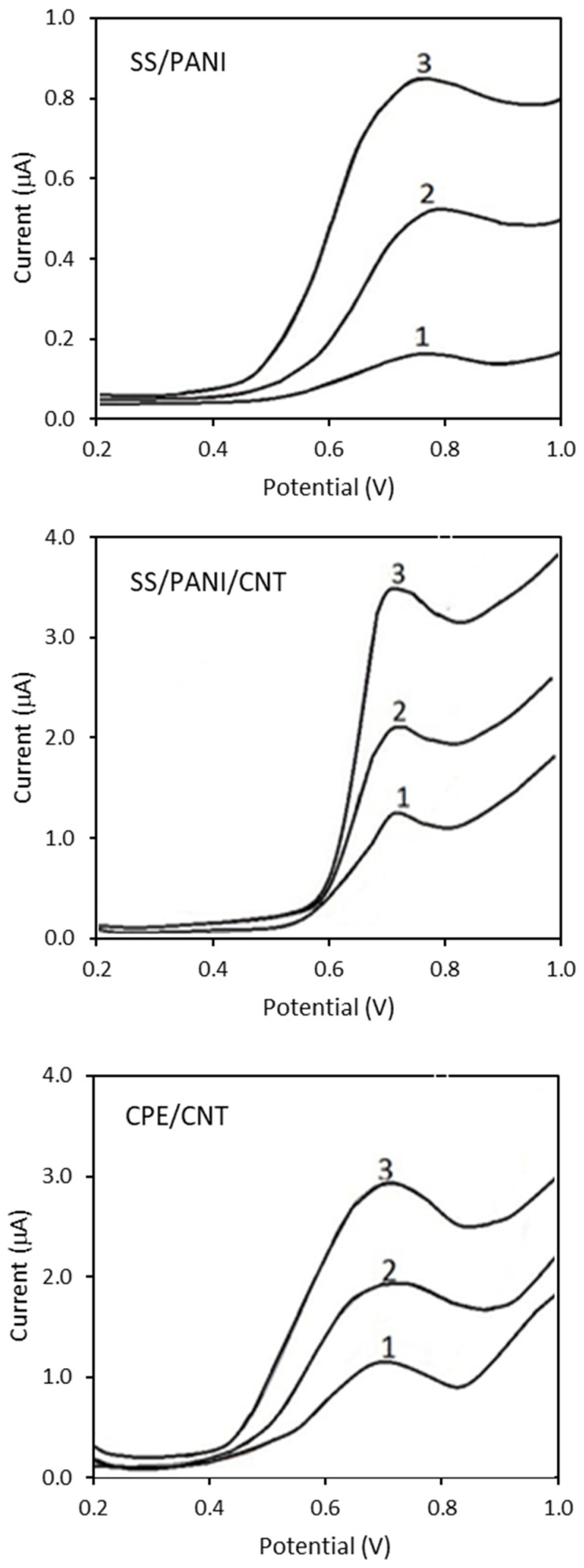
Differential pulse voltammograms registered for 0.5 mmol/L 4-CP in 0.1 mol/L solutions of Na_2_SO_4_ on SS/PANI and SS/PANI/CNT electrodes obtained after 1–10 cycles, 2–20 cycles, 3–30 cycles, as well as on CPEs modified with CNTs content of 1–2.5%, 2–5%, and 3–10%.

**Table 1 materials-15-03436-t001:** Physical properties of fibers based on SEM measurements.

Fiber	Diameter [µm]	Polymer Layer Thickness [µm]	Polymer Volume [mm^3^]
SS	247	-	-
SS/PANI/CNT 10 cycles	280	16.5	0.205
SS/PANI/CNT 20 cycles	300	26.5	0.342
SS/PANI/CNT 30 cycles	313	33.0	0.435

**Table 2 materials-15-03436-t002:** Comparison of linear correlation parameters and detection limits for 4-CP at different electrodes.

Electrode	*y* = a*x* + b	*R* ^2^	LOD	LOQ
			[µmol/L]	[µmol/L]
SS/PANI				
10 cycles	*y* = 0.37*x* − 0.001	0.992	7.30	24.3
20 cycles	*y* = 1.18*x* + 0.004	0.998	2.29	7.43
30 cycles	*y* = 1.57*x* − 0.005	0.996	1.72	5.73
SS/PANI/CNT				
10 cycles	*y* = 2.52*x* − 0.014	0.997	1.07	3.56
20 cycles	*y* = 4.24*x* − 0.020	0.999	0.64	2.13
30 cycles	*y* = 7.01*x* − 0.037	0.994	0.38	1.26
CPE/CNT				
2.5%	*y* = 2.38*x* + 0.004	0.998	1.13	3.76
5%	*y* = 3.94*x* − 0.013	0.997	0.68	2.26
10%	*y* = 5.94*x* − 0.040	0.995	0.45	1.50
bare/graphite electrode	*y* = 0.14*x* − 0.017	0.991	19.3	64.3

**Table 3 materials-15-03436-t003:** Comparison of recently reported electrochemical methods for the quantitative analysis of 4-CP.

Electrode	LOD [µmol/L]	Ref.
SS/PANI/CNT/30	0.38	This study
CPE/CNT/10	0.45	This study
SS/PANI/30	1.72	This study
montmorillonite modified CPE	0.02	[27]
blue lanthanide double-decker phthalocyanines modified GCE	0.04	[28]
MWCNT/gold nanoparticles nanocomposites modified GCE	0.11	[29]
molecularly imprinted polymer with PDDA-functionalized graphene	0.30	[30]
Ni(OH)_2_ nanoparticles-carbon nanotubes modified GCE	0.50	[31]
CPE modified by laccase immobilized on a hybrid nanocomposite	0.70	[32]
GO/NiO nanoparticle composite-ionic liquid modified CPE	0.70	[33]
SBA-15-NH_2_ modified CPE	1.40	[19]
Activated carbon modified CPE	2.38	[20]
CNT/Pt nanoparticles/rhodamine B modified GCE	3.70	[34]
hazelnut shell modified CPE	5.03	[22]
green lanthanide double-decker phthalocyanines modified GCE	6.14	[28]
halloysite/gelatin modified CPE	6.55	[21]
walnut shell modified CPE	7.90	[22]
expanded graphite-epoxy electrode	20.0	[35]
halloysite modified CPE	29.3	[21]

**Table 4 materials-15-03436-t004:** Intraday and interday assay reproducibility for the determination of 4-CP.

Electrode	Added 0.1 mmol/L			Added 0.5 mmol/L		
	Found ± SD [mmol/L]	CV [%]	Accuracy [%]	Found ± SD mmol/L	CV [%]	Accuracy [%]
Intraday						
SS/PANI/30	0.102 ± 0.006	6.25	101.91	0.501 ± 0.007	1.47	100.21
SS/PANI/CNT/30	0.101 ± 0.002	1.67	101.01	0.506 ± 0.005	0.59	101.23
CPE/CNT/10	0.098 ± 0.005	5.11	97.96	0.499 ± 0.001	0.16	99.95
Interday						
SS/PANI/30	0.097 ± 0.010	9.90	97.22	0.489 ± 0.011	2.97	97.98
SS/PANI/CNT/30	0.101 ± 0.008	8.33	100.84	0.508 ± 0.082	8.26	101.68
CPE/CNT/10	0.104 ± 0.006	5.80	104.50	0.498 ± 0.004	0.82	99.53

**Table 5 materials-15-03436-t005:** Results of 4-CP determination in spiked real samples.

Electrode	Tap Water Concentration Given—0.2 mmo/L			River Water (Vistula) Concentration Given—0.2 mmo/L		
	Found ± SD [mmol/L]	CV [%]	Recovery [%]	Found ± SD [mmol/L]	CV [%]	Recovery [%]
SS/PANI/30	0.195 ± 0.013	6.79	97.66	0.197 ± 0.013	6.45	98.61
SS/PANI/CNT/30	0.204 ± 0.013	6.28	102.24	0.190 ± 0.003	1.53	95.12
CPE/CNT/10	0.195 ± 0.004	1.93	97.72	0.192 ± 0.012	6.32	95.90

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
