# Peer review of "Electropolymerized Aniline-Based Stainless Steel Fiber Coatings Modified by Multi-Walled Carbon Nanotubes for Electroanalysis of 4-Chlorophenol"

_materials, 2022, doi:10.3390/ma15103436_

Round 1

Reviewer 1 Report

The authors presented a stainless steel fiber coated electropolymerized aniline without and with carbon nanotubes, as well as the carbon nanotubes modified carbon paste electrodes for electroanalysis of 4-chlorophenol. Although the manuscript is relatively well writing, its novelty is not enough. For example, to electropolymerize aniline on the fiber surface has been reported and the fabricated electropolymerized aniline-based fiber coating has been used to selectively extract phenols from water. The authors try to compare the effects of adding carbon nanotubes on the sensing performance but the significance is unclear. Additionally, because of the broad-spectrum of aniline-based fiber coating towards phenols, the selectivity of this sensor for the 4-CP is insufficient. The LOD of SS/PANI/CNT and CPE/CNT are comparable according to Table 2. But the selectivity of CPE/CNT should be much worse than SS/PANI/CNT. The purpose of the fiber design is not discussed enough in the manuscript.

Reviewer 2 Report

The author prepared electropolymerized aniline-based stainless steel fiber coatings. The results are interesting. However, some can be improved.

  1. The innovation of the manuscript should be clarified in the introduction more clearly. Only one sentence at the end was not enough.
  2. Some English expressions needs to be improved.
  3. "2.4. Carbon paste electrodes preparation" and "2.5. Carbon paste electrodes preparation " had the same titles. Please confirm them.

Reviewer 3 Report

The paper " Electropolymerized aniline-based stainless steel fiber coatings modified by multi-walled carbon nanotubes for electroanalysis of 4-chlorophenol" described the electropolymerized  aniline coatings  without and with carbon nanotubes. The topic is interesting, and the experiment is well supported the conlusions. The paper can be accepted after addressi the following comments.

  1. It is better to show the scale bars in the SEM images of Figure 4.
  2. Please cite more newly reference. The reference in the introduction part is too old.
  3. Please clarify clearly of curve round in Figure 3.
  4. Can you explain why the recovery of SS/PANI/CNT/30 is higher than 100% in Table 5?

Reviewer 4 Report

Stainless steel fiber-coated electropolymerized aniline with and without carbon nanotubes (SS/PANI and SS/PANI/CNT) were prepared and used for the detection of 4-chlorophenol based on differential pulse voltammetry.

A small addition of a modifier (CNTs) to a SS/PANI electrode increases peak currents. 

1. 6 mg CNT was added and the obtained suspension was mixed using ultrasound.

Question: What type of surfactant was used for the dispersion of CNTs?

2. The carbon paste electrode was prepared by thoroughly mixing graphite powder and paraffin oil (9:1 m/m) in a mortar with a pestle.

Question: Could you provide information related to the average particle size of graphite powder used?

3.  The increase of the modifier content yields an increase in the peak current. The results show that the modification of the used electrodes improves the efficiency of the chlorophenol electroanalysis.

Question: Could you provide data of CPE/PANi/CNT?

4. The capability of these stainless steel electrodes to detect 4-CP was compared with the CPEs modified by adding different amounts of CNTs (2.5% – 10%, m/m).

Question: What is the significant role of the stainless steel for the sensing system? Instead of the stainless steel wire, what kinds of wires can be used?

5. The SS/PANI fiber coatings modified by multi-walled carbon nanotubes
showed significantly greater sensitivity for the detection of 4-CP than the CPEs.

Question: Could you explain why the SS/PANi/MWCNT fibers showed the best performance in the viewpoints of electrical conductivity, electrochemical properties, wettability, and surface area to volume ratio? Also, I wonder if the MWCNTs used is better than SWCNTs for the sensing system.

6. The described methods using SS/PANI/30, SS/PANI/CNT/30, and CPE/CNT/10 electrodes have been applied to determine 4-CP in tap water and Vistula river water after the addition of 0.1 mol/L sodium sulfate.

Question: Please explain about the recovery difference between the tap water and river water (102.24% vs. 95.12% for SS/PANI/CNT30).

7.  The sensitivity of these methods is comparable to, or slightly worse, than those of electrochemical methods for 4-CP detection described by other authors.

Question: What is the main advantage of the SS/PANI/CNT fibers compared to the previous research system?

Round 2

Reviewer 1 Report

I still think this is a work without enough scientific advances to our current understanding, like a combination of A and B. If the authors try to improve your work, you need to investigate more parameters of this technology, for example, checking the selectivity of this fiber especially to other phenols in the environments, the linear response range, the stability etc. Only examining the spiked recovery by using the tap water or other aqueous environments is not enough to support that this is a sensing technology to 4-CP. According to my deduction, the film should be senstivie to a lot of phenols, which would introduce big interferences for the susbsequent electrochemical analysis. It means the analysis process will not work.
